# LEGO^®^-Based Therapy in School Settings for Social Behavior Stimulation in Children with Autism Spectrum Disorder: Comparing Peer-Mediated and Expert Intervention

**DOI:** 10.3390/brainsci14111114

**Published:** 2024-11-01

**Authors:** Luciana Oliveira Angelis, Fernanda Tebexreni Orsati, Maria Cristina Triguero Veloz Teixeira

**Affiliations:** 1Human Developmental Sciences Graduate Program, Mackenzie Presbyterian University (UPM), Sao Paulo 01302-907, SP, Brazil; 2Inclusive All Institute, Sao Paulo 04111-020, SP, Brazil; fernanda.orsati@gmail.com; 3Human Developmental Sciences Graduate Program, Center for Research on Childhood and Adolescence, Mackenzie Presbyterian University (UPM), Sao Paulo 01302-907, SP, Brazil; mctvteixeira@gmail.com

**Keywords:** autism spectrum disorder, LEGO^®^-based therapy, intervention, mental health, social skills, peer-mediated

## Abstract

Background: LEGO^®^-based therapy is a social development protocol that uses LEGO^®^ activities to support the development of a wide range of interaction skills, enhancing prosocial behaviors and mitigating the challenges associated with mental health difficulties and behavioral issues commonly observed in children with autism spectrum disorder (ASD). Objectives: This study aimed to explore the effects of LEGO^®^-based therapy on the social behavior and mental health of children with ASD, comparing stimulation mediated by expert and stimulation mediated by non-autistic peers. This study was approved by the Ethical Committee at Mackenzie Presbyterian University, ensuring adherence to ethical standards throughout the research process. Methods: This study involved 18 children with ASD, levels 1 or 2, with an intelligence quotient (IQ) above 70, and three typically developing peers, intelligence quotient (IQ) above 80, aged between 5 and 8 years old, of both sexes. Participants were randomized into three groups for stimulation (stimulation mediated by expert, by a non-autistic peer and control group). The measures were the Wechsler Abbreviated Scale of Intelligence, the Strengths and Difficulties Questionnaire (parent and teacher versions), the Inventory of Difficulties in Executive Functions, Regulation, and Aversion to Delay—Child Version, the Developmental Coordination Disorder Questionnaire, the Autism Behavior Checklist, and the Autistic Behavior Inventory. Results: After 14 sessions of 45 min in school settings, the participants of both groups (mediated by experts and non-autistic children) showed significant gains on social behavior. A statistically significant difference was observed between baseline sessions and probes (χ^2^ (5) = 25.905, *p* < 0.001). These gains were maintained in both follow-up points, 30 and 90 days after the completion of the stimulation sessions. Additionally, maladaptive behavior showed a significant decline when compared pre- and post-intervention. These improvements were sustained during follow-up assessments at 30 and 90 days. Conclusions: The results suggest that a structured intervention combined with peer-mediated stimulation may be an effective method for promoting adaptive and prosocial behaviors in children with ASD.

## 1. Introduction

Social communication differences and restricted and repetitive behaviors are the core characteristics of autism spectrum disorder (ASD) [1]. Furthermore, among the primary impairments observed in ASD are difficulties in social interaction, including challenges in understanding social cues, such as hand gestures, eye contact, and facial expressions, difficulties in sharing experiences and understanding others’ perspectives, engaging in joint attention, responding to others, taking turns, and participating in play [2].

The most recent statistics from the Centers for Disease Control and Prevention indicate that the prevalence of this condition is estimated to be 1 in 36 children [3]. In 2011 was conducted an important epidemiological study to stablish prevalence of ASD in Brazil showing rates of 27.2/10,000, a low frequency compared with prevalence data from developed countries in the same period [4]

Compared to non-autistic peers, children with ASD exhibit elevated levels of mental health issues, including emotional, behavioral, and attentional problems. It is suggested that some externalizing behaviors in ASD may be consequences of social difficulties, rather than a separate co-occurring disorder, particularly in high-functioning individuals [5,6]. Moreover, studies have highlighted these difficulties as barriers for school functioning and for the development of atypical social attention in individuals with ASD [7,8].

Widely used evidence-based interventions for ASD include behavioral therapies that focus on functional analysis, modeling and shaping behavior, naturalistic training, and parent training [9]. Additionally, developmental therapies that concentrate on social cognition skills, communication, social interaction, and peer-mediated stimulation are also recommended [10].

Stimulation through play can provide opportunities for developing social interaction skills (both initiating and responding to social interactions), expanding gestural and non-gestural communication, and expressing feelings and dissatisfactions, aiming for better quality of life and adaptive functioning [11,12]. The evidence reveals that prioritizing the stimulation of social interaction skills is one of the strategies for improving externalizing behavior problems in children with ASD [6]. The research on playful stimulation indicates that, when appropriately structured, children can benefit from the advantages that play offers in the development of social skills, emotional regulation, motor coordination, creativity, imagination, and adaptability [11,13]. However, for a child to benefit from play, they must be able to manage various sources of sensory stimulation, for example, simultaneously dealing with peers and the materials (toys).

One playful intervention that addresses social abilities, and both externalizing and internalizing behaviors with efficacy is LEGO^®^-based stimulation [14,15,16]. This approach has gained attention as a structured social development tool, particularly for children with ASD. The efficacy of this approach has been corroborated by a substantial body of research as an effective method for enhancing collaboration, communication, and social interaction between children [15,16]. It equips children with essential skills for interacting with others, including conversation and problem-solving abilities [14,15,16].

LEGO^®^-based stimulation provides structure and guidance, stimulating the underdeveloped skills of children with ASD [15,17,18,19]. It is possible for family members to be involved in the process [14,15,16,17,18,19]. It is also a playful activity that can be facilitated by non-autistic peers, which may enhance the quality of friendships [2]. LEGO^®^ therapy capitalizes on the interests of children with ASD as a motivating factor for learning and behavioral change [14,15,16].

Studies have emphasized that socially oriented therapies are effective when the child is exposed to non-autistic peers who can mediate and serve as models for learning, modeling, and generalizing. Peer-mediated interventions (PMIs) have shown significant positive effects on children with ASD in various settings [20,21].

PMI is an evidence-based practice that employs neurotypical peers as social models to encourage children with ASD to enhance their initiations, responses, and engagement, as well as to exemplify desired behaviors. The efficacy of PMI in enhancing social skills, social motivation, and overall social communication in children with ASD has been substantiated by empirical evidence [9,22]. It can be employed as a strategy to facilitate the development of interaction and social communication skills, Theory of Mind (ToM) skills, and play skills in children with ASD [21,23,24].

Interventions for ASD may draw upon a variety of evidence-based approaches to achieve a shared objective. This paper underscores the significance of social skills development in mitigating externalizing behaviors [6], the utility of a structured approach for children with ASD to benefit from play interventions [11], the effectiveness of LEGO^®^ therapy [15], and the evidence of peer mediation [20]. The integration of these approaches provides a more engaging and motivating learning environment for the child, thereby facilitating the development of specific social skills. Therefore, in Brazil there is still a need for explore the benefits of LEGO^®^-based therapy, particularly in scholar settings. This study aimed to explore the effects of LEGO^®^-based therapy on the social behavior and mental health of children with ASD, comparing stimulation mediated by an expert and stimulation mediated by non-autistic peers.

## 2. Materials and Methods

The design adopted for the study was a randomized controlled clinical trial with pre-intervention, post-intervention, and follow-up measurements carried out using 2 probes.

### 2.1. Participants

This study was conducted in the public-school district of Embu das Artes, São Paulo State, Brazil. For the recruitment of children with ASD, the Department of Special Education of the district invited 60 parents of children with ASD between 5 and 8 years. Of these 60 children, 32 parents agreed to participate in the study, but only 18 children met the inclusion criteria of having an intelligence quotient (IQ) above 70. The total study sample was composed of 64 participants, of whom 18 were children with ASD, 3 were non-autistic children, 21 were teachers, 21 were parents of the children, and 1 was a specialist. The sample was divided into three groups of six participants in each group. Children with ASD were randomly distributed among the three groups: (a) group with six children with ASD and one specialist (expert-mediated group), (b) group with six children with ASD and three non-autistic children (non-autistic-peer-mediated group), and (c) control group composed of six children with ASD. The children with ASD were assigned to expert-mediator and non-autistic peer groups in a 1:1:1 ratio. This study was approved by the Ethical Committee at Mackenzie Presbyterian University, ensuring adherence to ethical standards throughout the research process (CAAE: 57971422.2.0000.0084).

The inclusion criteria for the children with ASD were a diagnosis registered in the clinical report of the school, an IQ above 70, and not participating in other interventions to promote social skills and/or play therapy during the study. The inclusion criteria for the non-autistic children were an IQ equal to or greater than 80, an age between 5 and 8 years old, a normal classification on all scales of the Strengths and Difficulties Questionnaire (SDQ), and maximum scores on the pro-social behavior scale of SDQ (parent and teacher versions). Neurotypical peers were nominated by the teachers in the participating schools. The parents of the neurotypical peers authorized their participation in this study.

### 2.2. Measures

#### Instruments

(a)Wechsler Abbreviated Scale of Intelligence (WASI) [25,26] for children over 6 years old: This scale is composed of four subtests, which are Similarities, Vocabulary, Matrix Reasoning, and Cubes. It is a scale developed along the same lines as other traditional Wechsler scales, with the aim of providing three measures of intelligence—Verbal IQ, Execution IQ, and Total IQ. Currently, the WASI is standardized for the Brazilian population aged 6 to 89 years [25,26]. This scale was used to assess IQ in children aged 6 and over.(b)Non-verbal intelligence test SON-R 2½–7 [27] for children under 6 years old: It is a non-verbal tool that assesses broad areas of intelligence for children between 2½ and 7 years old, composed of four subtests, namely Mosaics, Categories, Situations, and Patterns, administered in this order. The test assesses spatial and visual–motor skills and abstract and concrete reasoning [27].(c)Sociodemographic questionnaire for sample characterization.(d)Strengths and Difficulties Questionnaire/parents and teacher versions (SDQ-P 4-17 and SDQ-T 4-17) [28]: It was used to identify social competencies and emotional and behavioral problems based on reports from parents or primary caregivers (SDQ-P 4-17) or teachers (SDQ-T 4-17). The SDQ items are distributed into five subscales of 5 items each: emotional symptoms, conduct problems, hyperactivity/inattention, peer relationship problems, and prosocial behavior, and the scores from the first four scales added together generate the total difficulties score. The SDQ calculation allows for classifying the score of each scale into the normal, borderline, and abnormal ranges. The instrument presents Brazilian evidence of content validity with cultural adaptation by Fleitlich, Cortázar, and Goodman [28].(e)Inventory of Difficulties in Executive Functions, Regulation, and Aversion to Delay—Child Version (IFERA-I) [29]: This was used to evaluate indicators of executive functioning in children with ASD. This can be answered by parents and teachers, and it is composed of 28 items divided into five subscales: Working Memory, Inhibitory Control, Flexibility, Delay Aversion, and Regulation. Each item is evaluated on a Likert scale with options “definitely not true”, “not true”, “partially true”, “true”, and “definitely true”, which are scored from 1 to 5, respectively. Higher scores are also indicative of greater difficulties.(f)Developmental Coordination Disorder Questionnaire (DCDQ-Brazil 3) [30]: This was used to identify motor coordination deficits. This questionnaire identifies motor coordination deficits. It was adapted for the Brazilian population by Prado and collaborators [30]. The questionnaire has 15 items, intended for parents to assess motor skills in everyday activities of their children. The instrument is divided into three groups: motor control, fine/writing motor skills, and general coordination. The scoring is a Likert scale, which ranges from a score of 1 (“not at all like your child”) to 5 (“extremely like your child”). The final score is the sum of the scores for each item, which varies from 15 to 75 points, with the total score classifying the child as “indicative or suspected of developmental coordination disorder (DCD)” or “Probably without DCD” according to three cutoff points of the age groups.(g)Autism Behavior Checklist (ABC) [31,32]: It was used to characterize the impairments of children with ASD based on the areas assessed by the subscales. The ABC is a list 57 atypical behaviors that are generally related to autism, grouped into five subscales that assess different areas: Sensory Stimulation (ES), Relationships (RE), Use of Body and Objects (CO), Language (LG), and Personal and Social Development (PS).(h)Session Satisfaction Form: This was developed by the researcher to evaluate the participant’s experience during each session. It consists of a coloring activity for the child to identify their feelings (happy or sad) at the end of each meeting.

The tests were administered by two experienced neuropsychologists, both doctoral students affiliated to the Human Developmental Science Graduate Program at Mackenzie Presbyterian University. These professionals conducted the assessments in accordance with rigorous scientific protocols, thereby ensuring the standardization and reliability of the data obtained.

During the first meeting, the project was explained in detail to the group of parents, materials were presented, and invitations were formalized. The informed-consent forms were signed by the parents and teachers, and the Assent Term and Image Use Term were signed by the parents and teachers of the children. After these procedures, the neuropsychologists administered the tests to the children.

In the second meeting, the teachers completed the SDQ (SDQ-T 4-17). Subsequently, the children were organized into their respective groups, and the first baseline data collection was conducted. In the third and fourth meetings, the second and third baseline data were collected from all participating children.

### 2.3. Procedure

The LEGO^®^-based therapy consisted of 14 sessions lasting 45 min each, twice a week, conducted in the special education support classroom for students with disabilities. All sessions were recorded using an iPhone positioned in the room and stored on a cloud-computing platform.

### 2.4. Setting and Materials

The location was prepared with visual resources developed by the researcher, containing systematic instructions on the stimulation dynamics, game rules, and guidelines on using LEGO^®^ materials to assist children during activities (Figure 1). The research employed the LEGO^®^ Early Simple Machines sets from the education line. The researcher developed the activities according to the participants’ age range, as defined by the LEGO^®^ manufacturer. Figure 1 shows the organization of the stimulation setting, with the arrangement of materials and participant positioning.

### 2.5. Baseline Data Collection

The initial baseline consisted of three sessions to verify the behavioral indicators studied. In this session, the researcher instructed the children to interact freely for about 10 min of play in pairs with the material.

### 2.6. Peer Training

The training was divided into three 30-min sessions and included teaching basic LEGO^®^ game rules, guidelines on how to encourage peers to provide tips, model, and reinforce behaviors through role-play and video modeling.

### 2.7. Intervention Sessions

All sessions were conducted in trios, following the same protocol for all groups, only the type of mediator varied. In the expert-mediated group, researchers conducted the stimulation; in the peer group, the neurotypical peers conducted play supervised by the researcher, following these stages: Stage 1 (approx. 2 min): greeting and model presentation; Stage 2 (approx. 3 min): defining the roles of each child for that session and handing out identification badges; Stage 3 (approx. 5 min): joint reading of the rules supported by visual aids for non-literate children to guide expected behaviors during activities; Stage 4 (approx. 20 min): trio model assembly structured by the digital step-by-step guide.

Three roles were played by all participants. The engineer observed the guide and explained to the supplier which pieces to separate and how they should be assembled. The supplier listened to the engineer’s instructions and, according to the request, separated and delivered the pieces to the builder, explaining how to assemble them. The builder received the pieces from the supplier and assembled the models according to the instructions received. Upon completing a step, the builder informed the team that they were ready for the next instructions. This step was repeated until the model was completed. Stage 5 (approx. 5 min): Session feedback where each child was encouraged to discuss their experience during the activity and record it. The researcher concluded the activities by placing a sticker on the child’s certificate according to rule compliance. Upon completing the stimulation sessions, the child received a participation certificate as a “LEGO^®^ Builder”; Stage 6 (approx. 5 min): At the end of the session, participants removed their badges and disassembled the model together, organizing the materials.

### 2.8. Reevaluation

The SDQ-P 4-17 was readministered for parents and SDQ-T 4-17 for teachers.

### 2.9. Follow-Up

During probe sessions, similarly to the baseline, participants were instructed to interact freely with the material, without the structured setting. The first probe was applied in the next session following the end of the protocol, the second 30 days later, and the third 90 days after the stimulation ended.

### 2.10. Control Group Stimulation

Children randomly chosen for this group continued with their usual activities and did not receive any special treatment. They remained in the Special Education Support Room and the Student Support Room for Students with Disabilities (SAED) in activities planned according to their Individual Education Plan (IEP) developed by the school. This group participated in this LEGO^®^-based therapy program after the project’s completion, following the last probe data collection.

### 2.11. Statistics

All analyses were conducted using JASP 0.18.3.0 software and were divided into two parts. Initially, descriptive statistics of the sociodemographic variables were performed to characterize the sample. A Kruskal–Wallis test was used to verify group equivalence at the pre-intervention stage and later to compare the three groups regarding the SDQ-P and SDQ-T variables (behavioral problems and pro-social behavior). Subsequently, the same variables were compared using a Wilcoxon test within the three separate groups (intragroup).

Behavioral data were obtained from analyzing 10-min excerpts of each session, selected by a random cut. Two trained psychologists coded the behaviors identified during the sessions using a registration protocol, analyzing the videos for each child separately in random order to minimize the chronological influence on result interpretation.

They had undergone systematic training prior to their involvement in the study. The training program comprised structured sessions that included the presentation and analysis of videos. During these sessions, the participants were instructed in detail about the criteria and protocols for evaluating behaviors. They also reviewed example videos, discussed their observations together, and received continuous feedback to refine their skills.

All evaluated behaviors were systematized and defined in a coding manual developed by the researcher to assist coders’ accuracy. This manual was explained in a face-to-face meeting to clarify any observer doubts. A 4-h training session using 3-min videos illustrating the studied behaviors (Offering Help, Asking for Help, Initiating Interaction, Responding to Interaction, and Execution) was conducted. During training, observers were instructed to record the timing and type of identified behavior, and after they reviewed each behavior in a second analysis to minimize errors. The training success criterion was set at an 80% agreement rate on responses issued during training with the researcher. When both observers achieved 80% accuracy in coding, they were authorized to analyze the collected research data.

In the data analysis phase, one observer analyzed 100% of the videos, while the second observer analyzed 20% of the same videos, aiming for an 80% agreement rate. Observer agreement was defined as both observers selecting the same category for each variable within a 3-s interval, calculated using the Kappa concordance coefficient.

For all listed behaviors, the number of occurrences per session was measured. Visual data analysis was performed through tables and graphical representations of the categories to identify changes in level, rate, or trend direction in the graph line. It aimed to demonstrate variations in stimulated behaviors throughout the intervention. Subsequently, group differences were calculated considering the mediator.

Intervention and probe graphs were subsequently generated to visually understand the number of behaviors observed at each analyzed moment, separating the expert and peer groups. A significance value of *p* ≤ 0.05 was adopted for statistical significance, with *p*-values of ≤0.06 and ≤0.07 considered statistically significant or marginally significant trends [33]. To understand the effect size of the differences, a Cohen’s d test was applied; for this test, the effect size was evaluated using the Cohen’s d test (0.20—small magnitude, 0.50—medium magnitude and 0.80—large magnitude).

## 3. Results

### 3.1. Description of Participants

A total of 21 children participated in this study, with three (14.29%) neurotypical children and 18 (85.71%) diagnosed with ASD. They were divided into three groups: six (33.3%) in the expert group, six (33.3%) peers, and six (33.3%) in the control group. In the expert group, the children’s ages ranged from six to eight years (M = 7.00; SD = 0.63), with one (5.6%) female and five (27.8%) males. In the peer group, ages ranged from five to eight years (M = 5.83; SD = 1.33), with five (27.8%) males and one (5.6%) female. In the control group, ages ranged from six to eight years (M = 7.50; SD = 0.83); all six were male (33.3%). For analysis, neurotypical children, who participated as mediators in the peer group, were excluded.

### 3.2. Group Equivalence in Pre-Intervention Evaluation

A Kruskal–Wallis test was performed to assess the equivalence between participants in the groups at the pre-intervention stage. The expert group had a mean IQ of 102 (SD = 17.0), while the peers had a mean IQ of 103 (SD = 9.54), and the controls had an IQ of 93.5 (SD = 13.9).

Regarding the scales applied in the pre-intervention assessment, the expert group scored an average of 90.0 (SD = 14.0) on the ICA, 42.8 (SD = 13.0) on the DCDQ, and 103 (SD = 11.1) on the IFERA. The peers scored 86.3 (SD = 33.6) on the ICA, 44.0 (SD = 8.72) on the DCDQ, and 109 (SD = 10.4) on the IFERA, while the controls scored 85.3 (SD = 7.74) on the ICA, 38.8 (SD = 15.8) on the DCDQ, and 106 (SD = 13.6) on the IFERA. 

Similarly, no statistically significant differences were found on the SDQ total scores completed by parents and teachers: SDQ total parents (H(2) = 0.831, *p* = 0.66), SDQ total teachers (H(2) = 2.919, *p* = 0.27), SDQ prosocial parents (H(2) = 3.350, *p* = 0.18), and SDQ prosocial teachers (H(2) = 1.442, *p* = 0.48).

There were no statistically significant differences between the groups regarding IQ, age, ICA, DCDQ, IFERA, and block/mosaic tests.

### 3.3. Differences in Mental Health Indicators Between Groups (Post-Intervention)

The Kruskal–Wallis test was conducted to assess differences in SDQ scores provided by parents post-intervention across the groups (expert, peer, and control), as shown in Table 1. The test was statistically significant (H (2) = 8.091, *p* = 0.017) for the SDQ-P total score.

Multiple comparisons demonstrated that the peer group had lower scores compared to the control group, with a large effect size (z = −9.222; *p* = 0.014, d = −2.01).

A statistically significant difference was found between the groups on the SDQ prosocial scores provided by parents (H (2) = 14.616, *p* = 0.001). Post hoc analyses indicated that the control group had lower prosocial behavior scores compared to the peer group, with a large effect size (z = 11.861; *p* < 0.001, d = −2.59).

For the SDQ total scores completed by teachers, the results were also statistically significant (H (2) = 11.635, *p* = 0.003). Group comparisons showed that the peer group had lower scores than the control group, with a large effect size (z = −11.111; *p* = 0.002, d = 2.42).

Finally, a statistically significant difference was found between the groups on the SDQ prosocial scores completed by teachers (H (2) = 12.286, *p* = 0.002). Pairwise comparisons suggested that the control group had lower prosocial behavior scores than the expert group, with a large effect size (z = 10.000; *p* = 0.013, d = −2.18), and lower scores compared to the peer group (z = 10.444; *p* = 0.003, d = −2.28).

### 3.4. Interobserver Agreement in Social Behaviors Analyses

Interobserver agreement was calculated based on visual analyses of sessions, coded from the identification of social behaviors promoted throughout the protocol: Initiate Interaction (II), Respond to Interaction (RI), Offer Help (OH), Request Help (RH), and Execution (E). The overall agreement was 86.3% for the total observed behaviors (n = 3430). 

The analysis was conducted using the Kappa coefficient, which indicated almost perfect agreement, with a Kappa value of 0.92 (95% CI 0.91–0.94).

### 3.5. Comparison of Total Behavior Means Between Groups Pre- and Post-Intervention

Before analysis, a Kruskal–Wallis test was conducted to assess differences in the average frequency of social behaviors before the intervention between the groups. No statistically significant results were found for the studied social behaviors: Initiate Interaction II (H(2) = 3.765, *p* = 0.15), RI (H(2) = 2.503, *p* = 0.28), RH (H(2) = 0.050, *p* = 0.97), OH (H(2) = 1.295, *p* = 0.52), E (H(2) = 2.000, *p* = 0.36), and total behaviors (H(2) = 3.781, *p* = 0.15). This analysis guaranteed that groups were comparable before intervention.

To access intervention effects regarding all social behaviors, mean differences were found between Baseline 1 and final probe (z = 3.398, *p* = 0.001, d = −5.10), follow-up 1 (z = 2.012, *p* = 0.047, d = −4.61), and follow-up 2 (z = 2.146, *p* = 0.035, d = −4.80), with a large effect size. The mean total of social behaviors in baseline 2 was lower than in the final probe (z = 2.817, *p* = 0.006, d = −5.13).

Table 2 presents the mean social behavior. A statistically significant difference was observed between the baseline sessions and the probes (χ^2^ (5) = 25.905, *p* < 0.001). Furthermore, the gains were maintained at both follow-up points, 30 and 90 days after the completion of the stimulation sessions.

As shown in Table 3, the average total social behaviors in baseline 3 was significantly lower than in the final probe (z = 4.292, *p* < 0.001; d = −5.31), follow-up 1 (z = 2.906, *p* = 0.005; d = −4.81), and follow-up 2 (z = 3.040, *p* = 0.003; d = −5.00), with a large effect size.

The mean for social behaviors by group at each time point (pre- and post-intervention) showed statistically significant differences. A chi-square test revealed a value of **χ^2^(5) = 25.905**, with ***p* < 0.001**, indicating significant variation between the groups (Figure 2).

### 3.6. Satisfaction of Participants

At the end of all sessions, children with ASD and non-autistic children completed an individual assessment of satisfaction. They were encouraged to describe their feelings about the session by selecting either a “fun” or “boring” image in a coloring activity. Out of a total of 210 evaluations (14 per child), 98.1% of the participants chose the “fun” option, indicating a high level of satisfaction, while 1.9% selected “boring”. These results showed a strong positive perception of the intervention with an overall satisfaction rate of 98.1% across all sessions (Figure 3).

## 4. Discussion

This study aimed to investigate the effectiveness of using a structured LEGO^®^ stimulation protocol to promote social behaviors and positive mental health indicators in children with ASD. To this end, a previously tested protocol [34] was replicated and expanded, comparing, in the present study, three different groups: expert-mediated, peer-mediated, and control group. In addition, this study assessed the maintenance of these gains through two subsequent follow-ups.

The primary objective was to assess the difference in mental health indicators of the participating children. Parents and teachers reported positive changes, with a reduction in difficulties on the following scales: Emotional Problems, Conduct Problems, Hyperactivity, and Peer Problems. The scores obtained reflected positively on the classification range of the participants’ mental health after the stimulation.

Therefore, intervention demonstrated statistically significant efficacy in both stimulated groups. In contrast, the control group, which did not receive the intervention, exhibited no improvement, underscoring the role of the intervention in promoting enhanced mental health and prosocial behaviors [20].

The positive outcomes observed in the intervention groups can be attributed to the structured nature of the program, which is primarily designed to enhance participants’ interaction skills and facilitate its implementation in the school environment. Previous studies indicate that psychosocial interventions aimed at developing social skills and managing challenging behaviors in mainstream schools have the potential to mitigate emotional and behavioral difficulties, thus promoting the well-being of these children [5,6,8]. Furthermore, research suggests that by improving the social functioning of children with ASD, there is a tendency to reduce behavioral problems, indicating that socialization is a critical target in interventions for this population [6,8].

Interaction with peers and activities conducted during the intervention provide opportunities for social learning, which can assist children in comprehending the expectations associated with social processes. This, in turn, can help to mitigate feelings of isolation and encourage the development of prosocial behaviors. Frequent interaction with peers creates an environment in which children with ASD can practice and reinforce adaptive social behaviors, markedly enhancing their mental health and overall well-being [22]. These findings underscore the efficacy of targeted interventions in promoting enhanced mental health outcomes.

This study verified whether the participating children exhibited enhanced social interaction patterns following structured stimulation with LEGO^®^ and whether the frequency of these behaviors increased over the course of the stimulation. This analysis was based on previous studies that employed similar methodological designs [35,36]. Our results showed that all children exhibited an increase in the number of interactions following 14 sessions of stimulation, with sustained gains observed at the 30- and 90-day marks, respectively, after the final session. These findings contribute to the growing body of evidence supporting the efficacy of the LEGO^®^-based therapy. This approach employs evidence-based behavioral strategies to teach social skills to participants, thereby creating an opportunity to facilitate the development of social skills with positive outcomes in the initial sessions [14,15,16]. The fact that the gains were maintained after the end of stimulation highlights the importance of a structured, play-based procedure that links social play to the interactional skills of children with ASD [37,38].

The structure and predictability of the environment showed support for cognitive rigidity, executive function impairments, and social cognition deficits. As recommended in a previous study [38], our approach promoted contextual understanding and the acquisition of social repertoire in participating children through imitation and modeling in a playful context. Our results were similar with Hu and colleagues’ (2018) [36] study conducted in an inclusive preschool in China, as well as Levy and Dunsmuir’s (2020) [14] research, in which the authors examined the effects of LEGO^®^ therapy on six male adolescents diagnosed with ASD (ages 11 to 14) who were stimulated during school hours in regular classrooms. Despite age differences, social interaction and cooperation improved among the participants in both studies.

The main findings of this study focus on the potential for non-autistic peers’ mediation. As the present study shows, peer-mediated interventions are frequently regarded as being as efficacious as those conducted by specialists, due to their capacity to engender naturalistic social interactions in a diversity of settings [21,23]. This approach capitalizes on the intrinsic social dynamics of peer relationships, which can facilitate a more accessible and less intimidating environment for children with ASD. Such activities encourage the practice of social skills in everyday situations and are considered more likely to generalize outside structured environments, as peers serve as models for appropriate behavior, thereby enhancing social learning [20,23]. Furthermore, they provide regular opportunities for interaction, often integrated into school and leisure activities, which facilitates consistent practice and reinforcement of social behaviors.

The evidence indicates that the involvement of peers is beneficial for children with ASD in terms of enhancing their communication and socialization skills, while also increasing their social motivation [21,23], as also shown in this work. Furthermore, the reinforcement elements inherent to peer-mediated interventions are more pronounced than those observed in adult-mediated interventions. Consequently, the naturalistic and integrated nature of peer mediation may contribute to the effectiveness of LEGO^®^-based therapy, which is a viable, low-cost, and low-training alternative to specialist-mediated interventions [21,36].

The results presented throughout the sessions indicate that, as proposed in the literature, LEGO^®^ materials are attractive to this population. The children reported a 98.1% satisfaction rate. This result supports the argument that LEGO^®^ is engaging for children with ASD [17,18,19,34]. Additionally, the well-defined tasks requested step-by-step offer comfort and encourage social interaction. It provides opportunities for experiences in situations that are similar to those encountered in everyday life, thereby assisting in the training of social behaviors that are required in other contexts, without the child feeling pressured [17,18,19,35,39,40,41,42,43,44].

The present study is in alignment with the findings of Gibson, Pritchard, and Lemos (2021) [37], which suggest that school settings are a promising space for stimulation. The findings suggest that LEGO^®^ intervention fosters peer collaboration, particularly in these settings. The authors posit that the school can serve as a promising space for promoting comprehensive child development.

Moreover, the improvements shown in peer engagement and better adaptive functioning, particularly in the social domain, may act as protective factors [14,37,45]. These findings are significant because these factors have been identified as a significant concern for children with ASD in the existing literature [5,46,47].

This research addressed two significant shortcomings in the existing literature on the subject. First, participants in both the peer-mediated and expert-mediated groups exhibited significant behavioral improvements at the 30- and 90-day follow-up assessments, emphasizing the potential of the school setting to facilitate lasting ASD outcomes. Moreover, the findings provide further support for the hypothesis that structured protocols employing evidence-based behavioral strategies, when applied appropriately and consistently, can suppress symptoms, and promote the acquisition of social skills and improve mental health.

The evidence indicating that expert-mediated mediation may yield superior benefits suggests that trained professionals should be involved in LEGO^®^ intervention programs to optimize outcomes. Nevertheless, the efficacy demonstrated by peer mediation suggests that this approach may offer a viable and potentially more accessible alternative in educational and therapeutic settings.

### Limitations and Future Research

The current study has several limitations. First, the sample was restricted to a small group of children with ASD in one public educational district, which makes it impossible to generalize the results to children with ASD in other environments or cultures. This study was conducted in a specific educational environment, which may be affected by school environment, teacher quality, and other factors, and the external validity of the research results may be limited to some extent. Future studies with larger samples and different contexts are recommended to improve the reliability of our results. The sample of children with ASD was not equally balanced between boys and girls. Furthermore, there was no control for the co-occurrence of psychiatric conditions in the ASD children.

## 5. Conclusions

The LEGO^®^-based therapy presents a promising and accessible approach for improving social development and addressing behavioral issues in children with ASD. By incorporating structured yet naturalistic interventions in familiar environments, such as schools, and leveraging peer mediation, this therapy fosters meaningful social interactions and facilitates the generalization of social skills. The results showed positive effects of LEGO^®^-based therapy on the social behavior and mental health of children with ASD, with greater benefits from the non-autistic-peers-mediated group. The consistent engagement in playful activities not only enhances social learning but also contributes to improved mental health outcomes by reducing social isolation and increasing motivation for participation. Given its potential to address key challenges faced by children with ASD, LEGO^®^-based therapy stands out as an effective intervention that merits further exploration and integration into broader therapeutic frameworks. These results highlight how this low-cost intervention has a positive impact on the behavior and mental health of the children with autism in this vulnerable population. Our findings are very promising and should be replicated in other middle- and low-income countries where children with autism do not have access to evidence-based therapies or therapies at all.

## Figures and Tables

**Figure 1 brainsci-14-01114-f001:**
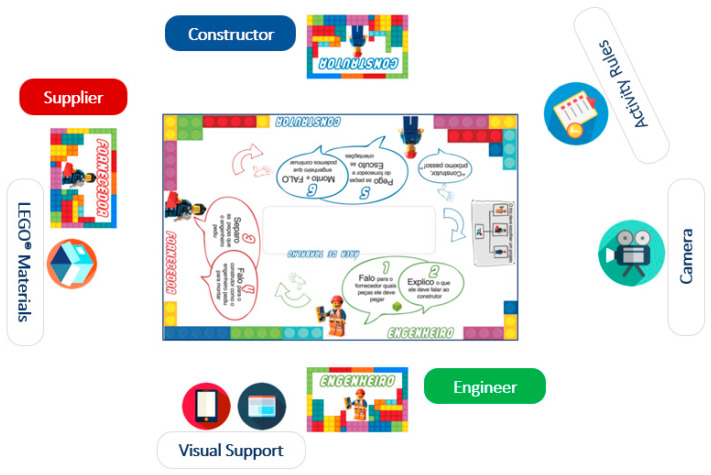
Stimulation setting created by the first author.

**Figure 2 brainsci-14-01114-f002:**
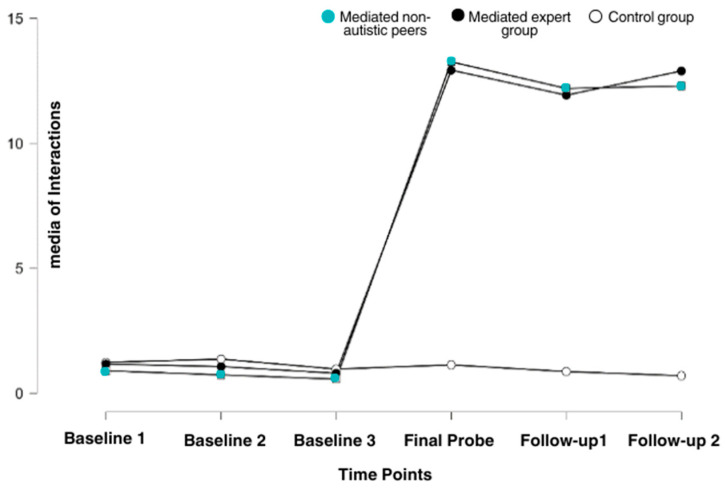
Comparison of social behaviors by group at each time point.

**Figure 3 brainsci-14-01114-f003:**
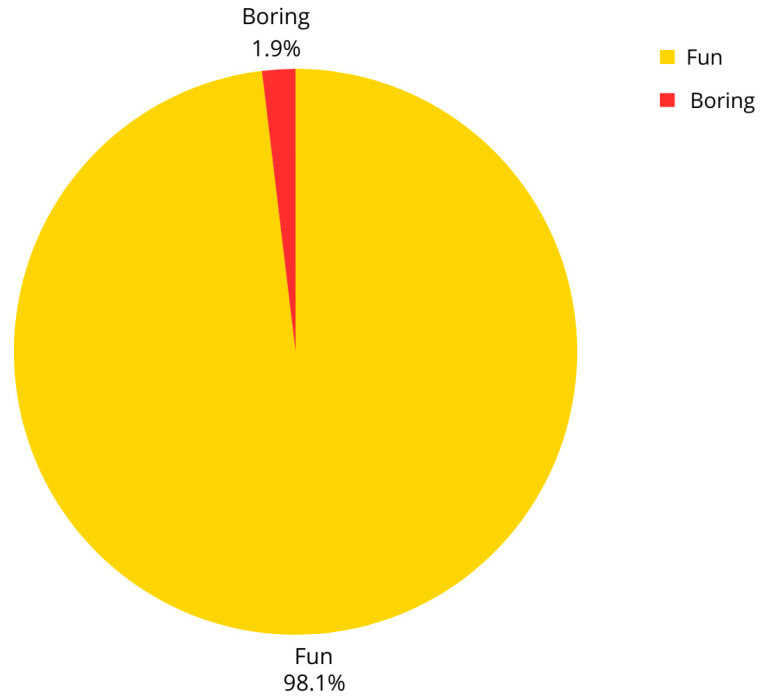
Index of satisfaction among the participants with the intervention.

**Table 1 brainsci-14-01114-t001:** Comparison of mean scores between groups on mental health indicators in pre- and post-intervention phases.

Scales	Groups	n	Mean (SD)	H	*p*	Mean (SD)	H	*p*
			Pre			Post		
SDQ total parents	Expert	6	21.00 (7.66)	0.831	0.66	13.5 (4.63)	8.091	0.017
	Peer	6	21.83 (4.75)			9.89 (7.14)		
	Control	6	21.83 (5.52)			22.3 (6.71)		
SDQ prosocial parents	Expert	6	5.00 (1.26)	3.350	0.18	9.17 (0.75)	14.616	0.001
	Peer	6	6.50 (1.22)			9.78 (0.44)		
	Control	6	5.33 (1.71)			4.67 (2.58)		
SDQ total teachers	Expert	6	13.50 (5.8)	2.619	0.27	9.17 (4.16)	11.635	0.003
	Peer	6	15.00 (7.9)			5.33 (4.58)		
	Control	6	18.50 (5.85)			18.1 (4.79)		
SDQ prosocial teachers	Expert	6	3.16 (1.94)	1.442	0.48	3.16 (1.94)	12.286	0.002
	Peer	6	4.50 (2.66)			4.50 (2.66)		
	Control	6	4.16 (0.98)			4.16 (0.98)		

**Table 2 brainsci-14-01114-t002:** Comparison of mean total social behaviors between baseline 1, baseline 2, baseline 3, final probe, follow up 1, and follow-up 2, by groups.

Time Points	Groups	n	Mean (SD)	χ^2^	*p*
Baseline 1	Expert	6	1.16 (0.95)	25.905	<0.001
	Non-autistic Peers	6	0.90 (0.27)		
	Control	6	1.23 (0.95)		
Baseline 2	Expert	6	1.06 (0.30)		
	Non-autistic Peers	6	0.73 (0.35)		
	Control	6	1.36 (0.23)		
Baseline 3	Expert	6	0.80 (0.40)		
	Non-autistic Peers	6	0.56 (0.44)		
	Control	6	0.96 (0.38)		
Final Probe	Expert	6	12.0 (0.66)		
	Non-autistic Peers	6	13.2 (3.31)		
	Control	6	1.13 (0.61)		
Follow-up 1	Expert	6	11.9 (1.56)		
	Non-autistic Peers	6	12.2 (2.30)		
	Control	6	0.86 (0.60)		
Follow-up 2	Expert	6	12.9 (3.44)		
	Non-autistic Peers	6	12.3 (3.11)		
	Control	6	0.70 (0.21)		

**Table 3 brainsci-14-01114-t003:** Multiple comparisons of total social behaviors between baseline 1, baseline 2, baseline 3, final probe, follow-up 1, and follow-up 2.

Time Points	Comparisons	z	*p*	d
Baseline 1	Baseline 2	0.581	0.563	0.02
	Baseline 3	0.894	0.374	0.20
	Final Probe	3.398	0.001	−5.10
	Follow-up 1	2.012	0.047	−4.61
	Follow-up 2	2.146	0.035	−4.80
Baseline 2	Baseline 3	1.476	0.144	0.17
	Final probe	2.817	0.006	−5.13
	Follow-up 1	1.431	0.156	−4.64
	Follow-up 2	1.565	0.121	−4.83
Baseline 3	Final probe	4.292	<0.001	−5.31
	Follow-up 1	2.906	0.005	−4.81
	Follow-up 2	3.040	0.003	−5.00
Final probe	Follow-up 1	1.386	0.169	0.49
	Follow-up 2	1.252	0.214	0.30
Follow-up 1	Follow-up 2	0.134	0.894	0.19

## Data Availability

The datasets used and analyzed during the current study are available from the corresponding author upon reasonable request. The datasets presented in this article are not readily available as they are part of an ongoing study and are still under analysis.

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
