# Peer review of "LEGO®-Based Therapy in School Settings for Social Behavior Stimulation in Children with Autism Spectrum Disorder: Comparing Peer-Mediated and Expert Intervention"

_brainsci, 2024, doi:10.3390/brainsci14111114_

Round 1
Reviewer 1 Report
Comments and Suggestions for Authors
General concept comments:
Comprehensive and well-described assessment instruments are used along with the purpose of their application. The intervention sessions are explained in detail, as well as the roles that each participant had to play. Scales have been applied that allow to know the pre-intervention situation among the different groups of participants. This allows to check that the results are reproducible based on the details given.
Specific comments:
Introduction: (Line 60) Add research references that corroborate LEGO based stimulation as an effective method for enhancing collaboration, communication and social interaction between children.
Materials and Methods- measures: (line 127) incorporate some information on the qualifications/specific training of the psychologist who administered the tests.
Materials and Methods- Statistics: (line 127) clarify the specific training of the two psychologists who coded the behaviors.
Results: There is no information or results on the feedback provided by each participant in each of the sessions (Stage 5).
Conclusions: I suggest including the main conclusions linked to the results obtained, in order to emphasise the contribution in each of the aspects assessed.
References: Of the total of 45 references, 15 belong to the last 5 years. No self-citation has been identified.
Author Response
Dear reviewer,
Thank you for taking the time to review our manuscript. Your feedback has been incredibly valuable in helping us improve our work. In response to the suggestions and comments provided, we have made the necessary revisions and would like to outline them below.
Comments1: Introduction: (Line 60) Add research references that corroborate LEGO based stimulation as an effective method for enhancing collaboration, communication and social interaction between children.
Response 1: We thank the reviewer for its careful reading of our work and great suggestions. The new references have been included to corroborate LEGO® based therapy as an effective method for enhancing collaboration, communication and social interaction between children (line 77)
Comments 2: Materials and Methods- measures: (line 127) incorporate some information on the qualifications/specific training of the psychologist who administered the tests.
Response 2: We thank the reviewer. The description of the qualification and process has been rewritten with more details in the Measures section (line 189 - 193).
Comments 3: Materials and Methods- Statistics: (line 127) clarify the specific training of the two psychologists who coded the behaviors.
Response 3: We thank the reviewer. The description of the training has been rewritten with more details in the Statistics section (line 275-279).
Comments 4: Results: There is no information or results on the feedback provided by each participant in each of the sessions (Stage 5).
Response 4: We thank the reviewer. We have included the data regarding satisfaction in numerical statistics in the Statistics section, item “3.5. Satisfaction of participants” (line 389 - 399). Additionally, there is a brief discussion on this topic in the Discussion section (line 475-478) .
Comments 5: Conclusions: I suggest including the main conclusions linked to the results obtained, in order to emphasise the contribution in each of the aspects assessed.
Response 5: The information was added to the in the Discussion section (line 456-464) and in the Conclusion section (line 520 - 531)
Comments 6: References: Of the total of 45 references, 15 belong to the last 5 years. No self-citation has been identified.
Response 6: We included into the references the only Brazilian pilot study conducted by the first author using the LEGO® Based Therapy (line 478)
We restructured the introduction in accordance with the reviewer's suggestions focusing on a new rationale and new studies.
We appreciate your time and consideration. We look forward to your feedback and any further suggestions to improve the manuscript.

Reviewer 2 Report
Comments and Suggestions for Authors
This study focuses on the effects of Lego therapy on the social behavior of children with autism, and compares the effects of expert intervention and peer mediated intervention, which has certain theoretical and practical significance. The research design is reasonable, the method is rigorous, and the results are persuasive. However, there are still some problems that need to be further improved and discussed.
1. When LEGO, ASD appears for the first time in the text, the full name should be indicated.
2. The study involves a total of 21 childern, including 18 children with autism, a relatively small sample size that may affect the generality of the findings. It is suggested that the sample size can be expanded in the follow-up study to improve the reliability of the research results.
3. The higher proportion of men in the expert group and the control group may have influenced the results. It is suggested that the gender ratio should be balanced as far as possible in sample selection.
4. The research is conducted in a specific school environment, which may be affected by school environment, teacher quality and other factors and the external validity of the research results may be limited to some extent. It is recommended that this issue should be discussed more fully in the discussion section and that suggestions be made that future studies can be validated in different environment.
5. The results showed that both the expert-mediated and peer-mediated intervention groups showed significant improvements in social behavior, while the control group did not. The authors need to discuss possible explanations for these results and compare them with the existing literature.
6. In the results and discussion section, the authors need to explain why peer-mediated interventions may be as effective as expert-mediated interventions, and the potential implications of this for future intervention strategies.
7. The writing quality of the article is generally good, but some parts may need further clarification and refinement. For example, the limitations of the study and future research directions could be discussed in more details.
8. The study design seemed reasonable, but the authors need to ensure that all participants had been adequately screened before the experiment to ensure they met the study's inclusion criteria.
9. The authors need to consider the generalizability of the study and whether the findings can be generalized to children with ASD in other environments or cultures.
10. In the abstract section of the article, the author did not clarify ethical issues in the design of the experiment.
11. The full-text citation of the article uses non-standard footnote notation.
12. The introduction does not elaborate on the research and background of ASD both domestically and internationally.
13. The paper reports significant improvements in the expert and peer groups on two outcome measures, along with a significant decline in maladaptive behavior post-intervention. The authors are advised to provide more comprehensive statistical data, such as effect sizes, confidence intervals, and the specific type of statistical tests used, to allow readers to fully assess the efficacy of the interventions.
14. It is suggested to discuss the potential applications of Lego® therapy in various environments (such as homes, community centers), and how to disseminate these findings to a wider population of individuals with ASD.
Comments on the Quality of English LanguageModerate editing of English language required.
Author Response
Dear reviewer,
Thank you for taking the time to review our manuscript. Your feedback has been incredibly valuable in helping us improve our work. In response to the suggestions and comments provided, we have made the necessary revisions and would like to outline them below.
This study focuses on the effects of LEGO® therapy on the social behavior of children with autism and compares the effects of expert intervention and peer mediated intervention, which has certain theoretical and practical significance. The research design is reasonable, the method is rigorous, and the results are persuasive. However, there are still some problems that need to be further improved and discussed.
Comments 1: When LEGO, ASD appears for the first time in the text, the full name should be indicated.
Response 1: We thank the reviewer. I would like to clarify that LEGO® is a registered trademark. The acronym ASD also was corrected (line 40)
Comments 2: The study involves a total of 21 children, including 18 children with autism, a relatively small sample size that may affect the generality of the findings. It is suggested that the sample size can be expanded in the follow-up study to improve the reliability of the research results.
Response 2: We thank the reviewer. This information was clarified in the section 2.1. Participants (line 110 - 133) and at Section 4.1 Limitations and future research (line 506-515)
Comments 3: The higher proportion of men in the expert group and the control group may have influenced the results. It is suggested that the gender ratio should be balanced as far as possible in sample selection.
Response 3: We thank the reviewer. The requested information has been included in the item 4.1 Limitations and future research (line 506-515)
Comments 4: The research is conducted in a specific school environment, which may be affected by school environment, teacher quality and other factors and the external validity of the research results may be limited to some extent. It is recommended that this issue should be discussed more fully in the discussion section and that suggestions be made that future studies can be validated in different environment.
Response 4: We thank the reviewer. The requested information has been included in the item 4.1 Limitations and future research (line 506-516)
Comments 5. The results showed that both the expert-mediated and peer-mediated intervention groups showed significant improvements in social behavior, while the control group did not. The authors need to discuss possible explanations for these results and compare them with the existing literature.
Response 5: We thank the reviewer. New aspects of the discussion were added to attend the recommendations in the discussion (line 417 - 425), and conclusion sections (line 516 - 532).
Comments 6. In the results and discussion section, the authors need to explain why peer-mediated interventions may be as effective as expert-mediated interventions, and the potential implications of this for future intervention strategies.
Response 6: We thank the reviewer.
We added new information to show how the peer-mediated interventions may be as effective as expert-mediated interventions in the Discussion section (line 457 - 475) and the potential implications of this for future intervention strategies was include in Conclusion section (line 516 – 532)
Comments 7. The writing quality of the article is generally good, but some parts may need further clarification and refinement. For example, the limitations of the study and future research directions could be discussed in more details.
Response 7: We thank the reviewer. We added new excerpts to improve the discussion section and added the limitations of the study and future research directions in more detail (line 506 - 515).
Comments 8. The study design seemed reasonable, but the authors need to ensure that all participants had been adequately screened before the experiment to ensure they met the study's inclusion criteria.
Response 8: We thank the reviewer. This information was detailed in section 2.1 - "Participants." (line 2.1) (line 110 – 133)
Comments 9. The authors need to consider the generalizability of the study and whether the findings can be generalized to children with ASD in other environments or cultures.
Response 9: We thank the reviewer. The requested information has been included in the item 4.1 Limitations and future research (line 506-515)
Comments 10. In the abstract section of the article, the author did not clarify ethical issues in the design of the experiment.
Response 10: We thank the reviewer. We added this information in the abstract (line 17) and in the methods section (line 123 - 125).
Comments 11: The full-text citation of the article uses non-standard footnote notation.
Response 11: We thank the reviewer. The information has been inserted according to the guidelines provided in the template of the journal (website).
Comments 12. The introduction does not elaborate on the research and background of ASD both domestically and internationally.
Response 12: We thank the reviewer. We added new excerpts in the introduction section to elaborate justification of the study using both domestically and internationally literature (line 45-49).
Comments 13. The paper reports significant improvements in the expert and peer groups on two outcome measures, along with a significant decline in maladaptive behavior post-intervention. The authors are advised to provide more comprehensive statistical data, such as effect sizes, confidence intervals, and the specific type of statistical tests used, to allow readers to fully assess the efficacy of the interventions.
Response 13: We thank the reviewer. This information is provided in section 3.3 Differences in Mental Health Indicators Between Groups (Post-Intervention) line (333-339)
Comments 14. It is suggested to discuss the potential applications of LEGO® therapy in various environments (such as homes, community centers), and how to disseminate these findings to a wider population of individuals with ASD.
Response 14: We thank the reviewer. The details have been included in section .1 Limitations and future research (line 506-516) and 5 Conclusion (line 516 - 532 ).
We appreciate your time and consideration. We look forward to your feedback and any further suggestions to improve the manuscript.

Round 2
Reviewer 2 Report
Comments and Suggestions for Authors
The manuscript can be accepted.
Comments on the Quality of English LanguageModerate editing of English language required.